# Glucose Oligosaccharide and Long-Chain Glucomannan Feed Additives Induce Enhanced Activation of Intraepithelial NK Cells and Relative Abundance of Commensal Lactic Acid Bacteria in Broiler Chickens

**DOI:** 10.3390/vetsci8060110

**Published:** 2021-06-12

**Authors:** Nathalie Meijerink, Jean E. de Oliveira, Daphne A. van Haarlem, Guilherme Hosotani, David M. Lamot, J. Arjan Stegeman, Victor P. M. G. Rutten, Christine A. Jansen

**Affiliations:** 1Department Biomolecular Health Sciences, Division Infectious Diseases and Immunology, Faculty of Veterinary Medicine, Utrecht University, 3584 CL Utrecht, The Netherlands; n.meijerink@uu.nl (N.M.); d.a.vanhaarlem@uu.nl (D.A.v.H.); v.rutten@uu.nl (V.P.M.G.R.); 2Cargill R&D Center Europe, B-1800 Vilvoorde, Belgium; jean_de_oliveira@cargill.com (J.E.d.O.); guilherme_hosotani@cargill.com (G.H.); 3Cargill Animal Nutrition and Health Innovation Center, 5334 LD Velddriel, The Netherlands; david_lamot@cargill.com; 4Department Population Health Sciences, Division Farm Animal Health, Faculty of Veterinary Medicine, Utrecht University, 3584 CL Utrecht, The Netherlands; j.a.stegeman@uu.nl; 5Department of Veterinary Tropical Diseases, Faculty of Veterinary Science, University of Pretoria, Onderstepoort, Pretoria 0110, South Africa

**Keywords:** broiler chickens, innate, adaptive, NK cells, T cells, IELs, microbiota, glucose oligosaccharide, long-chain glucomannan, in vitro-, in ovo-, in vivo screening

## Abstract

Restrictions on the use of antibiotics in the poultry industry stimulate the development of alternative nutritional solutions to maintain or improve poultry health. This requires more insight in the modulatory effects of feed additives on the immune system and microbiota composition. Compounds known to influence the innate immune system and microbiota composition were selected and screened in vitro, in ovo, and in vivo. Among all compounds, 57 enhanced NK cell activation, 56 increased phagocytosis, and 22 increased NO production of the macrophage cell line HD11 in vitro. Based on these results, availability and regulatory status, six compounds were selected for further analysis. None of these compounds showed negative effects on growth, hatchability, and feed conversion in in ovo and in vivo studies. Based on the most interesting numerical results and highest future potential feasibility, two compounds were analyzed further. Administration of glucose oligosaccharide and long-chain glucomannan in vivo both enhanced activation of intraepithelial NK cells and led to increased relative abundance of lactic acid bacteria (LAB) amongst ileum and ceca microbiota after seven days of supplementation. Positive correlations between NK cell subsets and activation, and relative abundance of LAB suggest the involvement of microbiota in the modulation of the function of intraepithelial NK cells. This study identifies glucose oligosaccharide and long-chain glucomannan supplementation as effective nutritional strategies to modulate the intestinal microbiota composition and strengthen the intraepithelial innate immune system.

## 1. Introduction

Restrictions on the use of antibiotics in the poultry industry encourage the development of new strategies to maintain or improve poultry health such as nutritional solutions [1,2]. Nutrients are digested in the intestine, a site of interplay between feed constituents, microbiota, and the immune system, which may all affect the development of the immune system and its function in chickens [3,4,5,6]. Nutritional modulation of innate immune responses is likely to be most beneficial for the health of chickens during the first week of life, since a high susceptibility to disease due to an immature adaptive immune system [7]. In addition, the intestinal microbiota is highly dynamic at this age, resulting in a higher susceptibility to intestinal infections, which emphasizes the importance of the innate immune system in young chickens [4]. Identification and use of feed additives that either modulate the immune system directly or indirectly through changes in the microbiota may contribute to improved resistance against pathogens during the early life of chickens.

Immediately post-hatch, the immune competence in chickens depends on maternal antibodies in addition to the innate immune system, of which natural killer (NK) cells and macrophages are key players [8,9]. The adaptive immune system is not fully developed upon hatch and functional T and B cell responses are only observed after approximately two to three weeks of life [8,9]. In previous studies in chickens, CD3^−^ IL-2Rα^+^ NK cells were identified as a major intraepithelial NK-subset in the intestine [6,10,11], and in addition, intraepithelial CD3^−^ 20E5^+^ NK cells were reported [6,11]. Furthermore, intraepithelial lymphocytes (IELs) also include high numbers of γδ T cells and cytotoxic CD8^+^ T cells [10]. Macrophages, B cells, and CD4^+^ T cells are located directly underneath the epithelium, the latter two mainly in Peyer’s patches [12,13]. Early feed modulation was shown before to increase resistance to pathogens by activation of macrophages and consequently subsequent adaptive reactivity [1,14,15]. Furthermore, both IL-2Rα^+^ and 20E5^+^ NK cells were shown to be important in the first response to viral [16,17,18] and *Salmonella enteritidis* [19] infections in chickens. 

Activation of chicken NK cells is characterized by increased surface expression of CD107 [20], and activation of macrophages may be assessed by the analysis of phagocytosis [21,22] and nitric oxide (NO) production [23]. Enhanced CD107 expression on NK cells is due to degranulation of vesicles containing perforin and granzyme, and results in apoptosis of the infected target cell [24]. Phagocytosis by macrophages leads to direct killing of internalized bacteria and antigen presentation to T cells [25], whereas NO production induces apoptosis in target cells, resulting in the killing of intracellular pathogens [26]. Analyzing the direct effects of feed constituents on the activation of NK cells and macrophages in vitro will provide preliminary information on the innate immune responsiveness in vivo. 

The development of the intestinal innate and adaptive immune system in chickens is influenced by exposure to microbiota, or their metabolites, and feed directly post-hatch [27]. Supplementation of probiotics like *Lactobacilli* and *Bifidobacteria*, and prebiotics like yeast and plant polysaccharides have been reported to lead to increased expression of genes implicated in innate and adaptive immune responses in chickens by binding to pathogen-associated molecular patterns (PAMPs) [28,29,30,31]. In addition, supplementation of plant polysaccharides affected numbers and cytotoxicity of NK cells, macrophages, T and B cells in chickens and humans [32,33,34,35,36,37,38,39,40,41,42]. Natural [43,44,45,46] and synthetic [47] compounds, and immunomodulatory drugs [48] have been shown to activate the innate and adaptive immune system, as demonstrated mainly by cytotoxicity assays. Despite many studies on immune modulation by feed compounds, their effects on NK cell subsets and activation have not been investigated in chickens before. Generation of more knowledge on modulatory properties of feed additives affecting innate immune cells will aid in finding feed strategies that increase immune responsiveness during the early life of chickens. 

Modulation of the intestinal microbiota through feed has been studied extensively in chickens [49,50], and positive effects on performance [51] and health [52] have been recognized. Pro- and prebiotics [53], plant and mushroom polysaccharides [32], and plant extracts [54] were shown to affect microbial composition and functionality. Nevertheless, combined analyses of feed additives affecting the microbiota and the function of innate immune cells are lacking in chickens.

In the present study, we investigated whether feed supplementation stimulated NK cells either directly or indirectly by modulation of the intestinal microbiota composition in young broiler chickens. A total of 69 potential feed additives were selected based on their characteristic of affecting the innate immune system, the intestinal microbiota, or both. The selection was narrowed down by assessing stimulatory properties on NK cells and macrophages in vitro, followed by assessing possible negative effects on embryonic development and hatch upon in ovo administration, and on performance traits upon in vivo administration. Based on these experiments, availability and regulatory aspects on use, glucose oligosaccharide, and long-chain glucomannan were subsequently investigated for their effect on innate and adaptive immune cells, and the composition of the intestinal microbiota. Our data identified glucose oligosaccharide and long-chain glucomannan as feed additives that positively affect NK cells, an important cell type of the innate immune system, and modulate the intestinal microbiota in broiler chickens. In this way, these feed additives may contribute to improved resistance to infections and hence the health of broiler chickens. 

## 2. Materials and Methods

### 2.1. Overall Experimental Design

A total of 69 compounds were selected based on their characteristic of affecting innate immunity, microbiota, or both, in chickens or other species either described in the literature or based on earlier studies performed by our industrial collaborator (Cargill Inc., Wayzata, MN, USA). Compounds were screened in vitro for activation of NK cells and macrophages and based on the observed effect on these innate immune cells, their availability in larger quantities and legally approved use in feed, six compounds were subjected to further analyses. Their impact on embryonic development and hatch were assessed following in ovo administration. In addition, their effect on performance traits upon feed supplementation directly post-hatch were studied. Based on the most interesting numerical results of these studies and highest potential to be produced in large scale, two polysaccharides (glucose oligosaccharide and long-chain glucomannan) were selected and supplemented to the diets of broiler chickens directly post-hatch to study their effects on NK cells and microbiota composition until three weeks of age. As glucose oligosaccharide and long-chain glucomannan were selected for testing in the final in vivo experiment, these compounds are highlighted in the in vitro screening alongside compounds showing contrasting effects.

### 2.2. Screening of the Effect of Compounds on Activation of NK Cells and Macrophages In Vitro

Selected compounds included plant extracts, fermentation products, vitamins, drugs, lipids, fungus extracts, polysaccharides, acid/salts, blend of essential oils and organic acids, yeasts, modified sugars, simple sugars, and emulsifiers (Appendix A). Powdered compounds were dissolved in DMSO (Sigma-Aldrich, Zwijndrecht, The Netherlands) before pre-dilutions were made at 10^3^ ppm/mL in complete RPMI medium (RPMI 1640 GlutaMAX-I supplemented with 10% heat-inactivated FCS and 50 U/mL penicillin/streptomycin; Gibco, United Kingdom), which were stored at 4 °C. DMSO diluted 1:5 in complete RPMI medium was used as the solvent control in all assays, which is equal to the highest amount of DMSO in the compound solutions. Compounds were added at concentrations of 10, 50, and 100 ppm and screened for their effect on NK cell activation using the CD107 assay. Possible effects on the macrophage cell line HD11 were determined by assessing phagocytosis and NO production. Each of the CD107, phagocytosis, and NO assays were performed in three independent experiments.

#### 2.2.1. NK Cell and T Cell Activation In Vitro and In Vivo as Assessed in the CD107 Assay

To study the possible effects of compounds on the activation of NK cells and cytotoxic CD8^+^ T cells, enhanced surface expression of CD107a as a consequence of degranulation was determined by flow cytometry [20]. For the in vitro screening, splenocytes of 14 day-old chicken embryos were isolated as previously described. In these spleens, a population of cells that resemble mammalian NK cells and lack surface expression of T or B cell-specific antigens is abundantly present [55], which are referred to as NK cells. ED14 NK cells were viably frozen and stored until use at −140 °C in complete IMDM medium (IMDM 2 mM Glutamax I supplemented with 8% heat-inactivated FCS (Lonza, The Netherlands), 2% heat-inactivated chicken serum, 100 U/mL penicillin, and 100 µg/mL streptomycin; Gibco) with 50% FCS and 10% DMSO, as described before [11]. Frozen ED14 NK cells were thawed and resuspended in complete IMDM medium at 4 × 10^6^ cells/mL.

Next, 1 × 10^6^ cells in 0.5 mL were incubated with 5 µL (10 ppm), 25 µL (50 ppm), and 50 µL (100 ppm) compound solutions in the presence of 1 µL/mL GolgiStop (Beckton Dickinson (BD) Biosciences, The Netherlands) and 0.5 µL/mL mouse-anti-chicken-CD107a-APC during 4 h at 37 °C, 5% CO_2_. Complete IMDM medium was used as the negative control, 50 µL of 1:5 DMSO as the solvent control, and a combination of 100 ng/mL phorbol 12-myristate 13-acetate (PMA, Sigma-Aldrich) and 500 ng/mL Ionomycin (Sigma-Aldrich) was used as the positive control. After incubation, cells were washed in PBA (PBS (Lonza) containing 0.5% bovine serum albumin and 0.1% sodium azide) and stained with the in vitro CD107 antibody panel (Table 1). Monoclonal antibodies anti-CD3 and -CD41/61 were used to exclude T cells and thrombocytes from the analyses of NK cell activation. Subsequently, cells were washed in PBS, stained with a viability dye, and fluorescence was analyzed by flow cytometry as described in Section 2.5.2. The fold change in CD107 expression upon incubation with the compounds was expressed relative to the negative control of each sample, which was set at 100%. 

In the in vivo experiment, cells were similarly incubated in the presence of Golgistop and CD107a-APC and stained with the in vivo CD107 antibody panel (Table 1). In addition to NK cell activation, T cell activation was analyzed in cells expressing CD3 and CD8α. The percentage of CD107 expression was determined within the total NK or CD8α^+^ T cell population. 

#### 2.2.2. Assessment of the Effect of Compounds on the Phagocytic Activity of the Macrophage Cell Line HD11 In Vitro

To determine the effects of the compounds on the activation of macrophage-like HD11 cells, a phagocytosis assay was performed. The chicken HD11 cell line [56], stored at −140 °C in complete RPMI medium with 50% FCS and 10% DMSO, was thawed and used after 3 to 20 passages. HD11 cells were cultured in complete RPMI medium in 75-cm^2^ cell culture flasks (Corning B.V., Amsterdam, The Netherlands) at 37 °C, 5% CO_2_, and passaged twice every week. HD11 cells were harvested from cell culture flasks when cells were at ~90% confluency, using a 0.05% trypsin/EDTA solution (Gibco). Subsequently, HD11 cells were counted and resuspended in complete RPMI medium at a concentration of 2 × 10^5^ cells/mL. Cells were seeded at 1 mL/well in 24-well cell culture plates (Corning Costar, Amsterdam, The Netherlands) and cultured overnight at 37 °C and 5% CO_2_. After culture, HD11 cells were incubated with 10 (10 ppm), 50 (50 ppm), and 100 (100 ppm) µL/well of compound solutions and incubated for 24 h at 37 °C and 5% CO_2_. As described previously, 1 μm crimson red-fluorescent beads (carboxylate-modified FluoSpheres, Invitrogen, The Netherlands) were used as targets for phagocytosis [21,22]. After 24 h of incubation, 1 × 10^7^ LPS (Sigma-Aldrich)-coated beads [21] were added to the wells with compound-incubated cells and wells were incubated for another 4 h at 37 °C, 5% CO_2_ to allow the cells to engulf the beads. The controls included incubation with 100 µL/well of complete RPMI medium followed by adding 1 × 10^7^ uncoupled beads as the negative control to determine baseline phagocytosis and 100 µL/well of 1:5 DMSO, followed by the addition of 1 × 10^7^ LPS-coated beads as the solvent control. Furthermore, incubation with 100 µL/well of complete RPMI medium followed by adding 1 × 10^7^ LPS-coated beads was used as the reference control and 100 µL/well of complete RPMI medium followed by the addition of 1 × 10^7^ IgY (Agrisera AB, Uppsala, Sweden)-opsonized beads [22] as the positive control to determine the highest level of phagocytosis. After the 4 h of incubation, supernatants were harvested, and the adherent cells were washed twice in PBS at room temperature (RT), followed by harvesting the cells using warm 5 mM, pH8 UltraPure EDTA (Sigma-Aldrich). HD11 cells were transferred to 96-well U-bottom plates (Greiner Bio-One B.V., Alphen aan den Rijn, The Netherlands), and subsequently washed in PBS, stained for viability, and analyzed by flow cytometry as described in Section 2.5.2. The percentage of total bead uptake was determined in viable HD11 cells using the gating strategy described in De Geus et al., (2012), which included consecutive selection of the HD11 cell population, viable cells, and total amount of beads. The fold change of bead uptake upon incubation with the compounds was expressed relative to the bead uptake of LPS-coated beads in unstimulated cells. This reference control was set at 100%.

#### 2.2.3. NO Assay to Assess Activation of the Macrophage Cell Line HD11 In Vitro

Possible effects of compounds on nitric oxide (NO) production by HD11 cells were measured by the Griess assay conducted on the culture supernatant [23]. HD11 cells were cultured, harvested, and resuspended in complete RPMI medium at 2 × 10^5^ cells/mL as described for the phagocytosis assay in Section 2.2.2. Cells were seeded at 1 mL/well in 24-well cell culture plates (Corning Costar, Corning Life Sciences B.V., Amsterdam, The Netherlands) and cultured overnight at 37 °C and 5% CO_2_. Next, the HD11 cells were incubated with 10 (10 ppm), 50 (50 ppm), and 100 (100 ppm) µL/well of compound solutions and 100 µL/well of complete RPMI medium was added as the negative control, 100 µL/well of 1:5 DMSO as the solvent control, and 100 ng/mL of lipopolysaccharides (LPS, Sigma-Aldrich) targeting *E. coli* O127:B8 as the positive control. After 48 h incubation at 37 °C and 5% CO_2_, 50 μL of supernatant was harvested and transferred to three individual wells of a 96-well flat-bottom plate (Corning Costar, Amsterdam, The Netherlands) for measurement of the nitrite concentration. A 3.13–200 μM nitrite (NaNO_2_) standard dilution series (Sigma-Aldrich, Zwijndrecht, The Netherlands) was included to generate a standard curve. Griess assay reagents were made by dissolving N-(1-naphtyl) ethylenediamine at 3 g/L and sulfanilamide at 10 g/l (both from Sigma-Aldrich) in 2.5% phosphoric acid (Supelco, Merck, MO, USA). These reagents were mixed 1:1 and 50 μL was added to the wells with cell culture supernatants and standards. The plate was then gently shaken in the dark at RT for 10 min at 700 rpm on a plate shaker (Schüttler MTS 4, IKA, Wilhelmshaven, Germany). The Griess reagents mixture turned purple upon reaction with nitrite ions in the cell culture supernatant. Finally, the optical density (OD) of each well was measured at 550 nm using a FLUOstar Omega microplate reader (BMG Labtech, Ortenberg, Germany) and the nitrite concentration of each sample was determined according to the nitrite standard curve.

### 2.3. Screening of the Effect of Compounds Following In Ovo Administration 

Possible negative effects of the compounds on embryonic growth, hatchability, and the number of peripheral blood cells were determined. A commercial hatchery (Morren B.V., Lunteren, The Netherlands) supplied 1200 Ross 308 14-day old embryonated eggs, they were weighed and incubated under optimal conditions in the experimental hatchery at the Cargill Animal Nutrition Innovation Center (Velddriel, The Netherlands). At embryonic day (ED) 15, eggs were randomly distributed in a complete randomized block design with six blocks including 20 treatment groups per block and 10 eggs per treatment in each block (replicates). At ED18, eggs were disinfected with alcohol spray, followed by injection of 1 mL of the respective compound solutions into the amniotic fluid using a 23-gauge disposable needle according to Tako et al., (2004). NaCl saline solutions (0.4%) contained 0.02%, 0.2%, and 2% of feed additive for the polysaccharides (P1, P2, P8), simple sugar (SS2), modified sugar (MS3), and 0.0015 mg, 0.0030 mg, and 0.0060 mg for the lipid (L1). Eggs that were not injected acted as the negative control group and 0.4% NaCl saline solution was injected as the solvent control. At ED19 and ED20, weights of eggs in the different groups were recorded and twelve eggs per group (two per replicate) were randomly selected and sacrificed to determine the weights of the embryo, yolk, and liver to assess embryonic growth by the use of yolk reserves. The relative embryo, yolk, and liver weights were expressed as a percentage of egg weight and used to determine the ratios between relative yolk and liver weight as a parameter of embryonic growth. At ED21 of the hatching period, twelve chicks per group (two per replicate) were randomly selected, weighed, and sacrificed to record weights of the chick, and remaining yolk and liver to assess embryonic growth. Numbers of internally or externally cracked eggs and hatched chicks were recorded and used to determine percentage hatchability. Numbers of unhatched eggs were counted, opened, and classified as dead embryos. In addition, four hatched chicks per group from different blocks were randomly chosen and sacrificed, and blood (~1 mL) was collected in EDTA tubes (VACUETTE^®^ K3E EDTA, Greiner Bio-One, Alphen aan den Rijn, The Netherlands) to determine white blood cell counts. One in ovo screening was performed. The number of replicates per treatment within the in ovo experiment was based on power analysis, which was adjusted for the specific facility and aligned to the 4Rs to reduce the use of animals in research by having a solid experimental design. For these calculations, we relied on historic data, obtained within the same experimental facility and with similar type of nutritional interventions.

The numbers of lymphocyte subsets were determined as described previously [57] by staining peripheral blood using BD Trucount™ Tubes (BD Biosciences, Vianen, The Netherlands) according to the manufacturer’s instructions. First, 200 µL of whole blood was fixed by mixing thoroughly with 40 µL Transfix^®^ (Thermo Fisher Scientific, Nieuwegein, The Netherlands), and subsequently diluted 1:50 in PBA. Next, 20 µL of the monoclonal antibody mix (Table 1) was added to BD Trucount Tubes followed by 50 µL of diluted blood, and this mixture was incubated for 15 min in the dark at RT. Subsequently, 450 µL PBA was added to the tubes, and cells and beads were measured by flow cytometry (FACSCANTO II Flowcytometer, BD Biosciences, Vianen, The Netherlands). Per sample, 20,000 beads were collected and data were analyzed with FlowJo software (FlowJo LCC, BD Biosciences, Ashland, OR, USA). According to the manufacturer’s instructions, cells positive for CD45 (lymphocytes) were gated and within this gate, CD3 negative and BU-1 negative cells (NK cells), BU-1 positive cells (B cells), and CD3 positive cells (T cells) were determined. The absolute number of immune cells per µL blood was calculated using the following formula: (number positive cell events/20,000 (number of beads)) * (48,550 (number of beads per test)/1 (test volume)). 

### 2.4. Screening of the Effect of Compounds on Growth Performance In Vivo

For the in vivo screening of possible effects on performance traits, 720 one-day-old Ross 308 broiler chicks were obtained from a commercial hatchery (Welp Hatchery, Bancroft, IA, USA), weighed, and randomly distributed in a complete randomized block design with 12 blocks including 12 dietary groups per block and five chickens per group in each block (replicates) at the Cargill Animal Nutrition Innovation Center (Elk River, MN, USA). Chicks were allocated into 144 pens, received water and standard or compound-supplemented commercial starter and grower feeds ad libitum. The 12 diets included standard diet (negative control) or feed supplemented with either of the selected compounds: 0.0625% *Saccharomyces cerevisiae* fermentation product (SCFP, positive control; XPC Ultra^TM^, Diamond V, Cedar Rapids, IA, USA), polysaccharides P1 (0.02%, 0.2%), P2 (0.2%), P5 (0.02%), P8 (0.02%, 0.2%), simple sugars SS2 (0.2%), SS2-S (0.02%, 0.2%, Sigma-Aldrich), and modified sugar MS3 (0.02%), respectively. Per treatment group (*n* = 60), chickens were weighed at a regular interval and feed intake was measured. Body weights and feed intake were used to calculate average daily gain and feed conversion ratio (FCR), respectively, at days 0, 7, 13, and 21. FCR was calculated by dividing the amount of feed uptake by the growth of the chickens at 0 to 7, 7 to 14, and 14 to 21 days of age. Housing temperature, ventilation, and lighting were according to recommendations for breeder chickens (Aviagen, 2018) and industry standards. All procedures were approved by the Animal Care and Use Committee of Cargill (Wayzata, MN, USA). The in vivo screening was performed once, and the number of replicates incorporated were based on power analysis. The power analysis was adjusted for the specific facility and aligned to the 4Rs to reduce the use of animals in research by having a solid experimental design. For these calculations, we relied on historic data, obtained within the same experimental facility and with similar type of nutritional interventions.

### 2.5. In Vivo Supplementation of Glucose Oligosaccharide and Long-Chain Glucomannan and Their Modulatory Properties on Immune Cells and Microbiota Composition

For the subsequent in vivo experiment studying the effect of glucose oligosaccharide and long-chain glucomannan on NK cells and microbiota composition, 15 sixteen-, seventeen-, and eighteen-day old embryonated eggs were obtained from a single Ross 308 broiler breeder flock at a commercial hatchery (Lagerwey, Barneveld, The Netherlands). Eggs were disinfected with 3% hydrogen peroxide and placed in disinfected egg hatchers in one stable at the facilities of the Department of Population Health Sciences, Faculty of Veterinary Medicine, Utrecht University, The Netherlands. The treatment of eggs with a low concentration of hydrogen peroxide does not influence the intestinal microbiota composition, since the development of the intestinal microbiota was not affected by this treatment in a previous study [6]. Directly upon hatch, chickens were weighed, labelled, and housed in one of the three different floor pens according to their feed group, with a solid wall separating the pens. Pens were lined with wood shavings (2 kg/m^2^), and water and standard or compound-supplemented commercial starter and grower feeds were provided ad libitum (Research Diet Services, Wijk bij Duurstede, The Netherlands). A standard lighting and temperature scheme for Ross broiler chickens was used. The animal experiments were approved by the Dutch Central Authority for Scientific Procedures on Animals and the Animal Experiments Committee (registration number AVD1080020174425) of Utrecht University (The Netherlands) and all procedures were done in full compliance with all relevant legislation. The in vivo experiment was performed once in which the number of chickens included per group was based on power analysis. 

Before the start of the experiment, environmental swabs (FLOQSwabs^®^, COPAN, Brescia, Italy) were taken of the hatchers and floor pens before hatching as well as of the hatchers after hatching. Swabs were stored at RT in 0.5 mL DNA/RNA Shield (Zymo Research, Irvine, CA, USA) to determine the microbiota composition of the environment. Directly post-hatch, chicks (*n* = 15) were provided with standard diet (control) or feed supplemented with 0.2% of either glucose oligosaccharide (P1) referred to as feed 1 (F1), or long-chain glucomannan (P2) referred to as feed 2 (F2). At days 7, 14, and 21, five chickens per feed group were randomly selected and sacrificed to collect ileum tissue (±10 cm distal from Meckel’s diverticulum), spleen, and contents of ileum (distal from Meckel’s diverticulum) and ceca for immunology and microbiota analyses. To calculate absolute cell numbers, ileum segments and spleens were weighed immediately after collection of the tissues, prior to isolation of cells. After isolation, cell numbers in the resulting suspension were calculated. This resulted in the total cell number, expressed as IELs per mg ileum or leukocytes per mg spleen. To calculate the absolute numbers of NK and T cells within the live IEL or leukocyte populations, the percentages of cells positive for the markers expressed on these cell types were used, which were determined in the flow cytometry analyses. Absolute cell numbers were calculated using the following formula: (absolute number IELs or leukocytes per mg tissue) × (percentage positive cells in the gate of interest of the live lymphocyte population). Intestinal contents were collected using sterile plastic cell culture loops, subsequently transferred into 2 mL sterile tubes containing 0.5 mL DNA/RNA Shield (Zymo Research), and stored at RT for DNA extraction. All chickens were weighed prior to post-mortem analyses to calculate body weight gain over the previous feeding period.

#### 2.5.1. Isolation of Immune Cells from the Intestine and the Spleen

The procedure to isolate IELs was performed as described previously and does not result in contamination with immune cells from the lamina propria [11]. Ileum segments were washed with PBS to remove the contents, cut in sections of 1 cm^2^, and washed again. Subsequently, the IELs were collected by incubating the sections for 15 min in a shaking incubator at 200 rpm at 37 °C in EDTA-medium (HBSS 1x (Gibco) supplemented with 10% heat-inactivated FCS (Lonza) and 1% 0.5M EDTA (UltraPure™, Invitrogen, Nieuwegein, The Netherlands)) after which the supernatants were harvested. This procedure was repeated three times using the remaining tissue sections. Subsequent supernatants containing the IELs were pooled and centrifuged for 5 min at 1200 rpm at 20 °C (Allegra™ X-12R Centrifuge, Beckman Coulter, The Netherlands). Cells were then resuspended in PBS, IELs were isolated using Ficoll–Paque Plus (GE Healthcare, Hoevelaken, The Netherlands) density gradient centrifugation for 12 min at 1700 rpm at 20 °C, washed in PBS by centrifugation for 5 min at 1300 rpm at 4 °C, and resuspended at 4.0 × 10^6^ cells/mL in complete IMDM medium. Spleens were homogenized using a 70 µm cell strainer (BD Biosciences, Vianen, The Netherlands) to obtain a single cell suspension. Next, leukocytes were isolated by Ficoll–Paque density gradient centrifugation (20 min, 2200 rpm, 20 °C), washed in PBS, and resuspended at 4.0 × 10^6^ cells/mL in complete IMDM medium as described for IELs. Cell suspensions were analyzed for subsets and activation of NK and T cells as described in Section 2.5.2 and Section 2.2.1, respectively.

#### 2.5.2. Phenotypic Characterization of IELs and Splenic Leukocytes by Flow Cytometry

Presence of NK and T cell subsets were determined among IELs and splenic leukocytes at 7, 14, and 21 days of age as described previously [11]. Cell populations (1 × 10^6^) were stained with a panel of antibodies specific for surface markers known to be expressed on NK cells as well as with anti-CD3 to exclude T cells from analysis. In addition, cells were stained with a panel of antibodies specific for surface markers that distinguish γδ T cell- and cytotoxic CD8^+^ T cell-subsets (Table 1). Staining with primary and secondary antibodies was performed in 50 µL PBA. Cells were incubated for 20 min at 4 °C in the dark, washed twice by centrifugation for 5 min at 1300 rpm at 4 °C in PBA, after primary staining, and in PBS after secondary staining. Subsequently, to be able to exclude dead cells from analysis, cells were stained in 100 µL PBS with a viability dye (Zombie Aqua™ Fixable Viability Kit, Biolegend, The Netherlands) for 15 min at RT in the dark, washed twice in PBA, and resuspended in 200 µL PBA. Of each sample, either 150 µL or a maximum of 1 × 10^6^ viable cells were analyzed using a CytoFLEX LX Flow Cytometer (Beckman Coulter), and data were analyzed with FlowJo software (FlowJo LCC, BD Biosciences, Ashland, OR, USA). 

The gating strategies used to analyze NK cells, γδ T cells, and cytotoxic CD8^+^ T cells were described previously [6,19]. In short, gating included consecutive selection for lymphocytes (FSC-A vs. SSC-A), singlets (FSC-A vs. FSC-H), viable cells (Live/Dead marker-negative), followed by the selection of specific cellular subsets and upregulation of the activation marker CD107 according to the staining panels. NK cell subsets were gated on CD3^−^ cells expressing either IL-2Rα or 20E5. To assess NK cell activation, CD3^−^CD41/61^−^ cells were gated within the live cells and expression of CD107 within this subset was assessed. T cell subsets were gated by selecting CD3^+^CD4^−^ cells that were positive for TCRγδ (γδ) or negative (CD8^+^ αβ) and subsequently, expression of CD8αα and CD8αβ within both γδ and CD8^+^ αβ T cells was assessed. To assess T cell activation, CD3^+^CD41/61^−^ were gated within the live cells and subsequently CD8α^+^ T cells were selected in which expression of CD107 was assessed.

#### 2.5.3. Microbiota Composition of Ileum and Ceca

DNA was purified from ileal and cecal samples using the ZymoBIOMICS DNA Kit according to the manufacturer’s instructions (Zymo Research, Irvine, CA, USA). Bacterial 16S ribosomal RNA genes were then amplified by running one PCR cycle while incorporating a cy-5 fluorescent labeled nucleotide, as described previously for labeling samples in microarray analysis [58]. Labeled PCR amplicons were then hybridized on a microarray chip containing probes for intestinal bacteria previously selected as biomarkers for broiler performance and intestinal health [58]. Microarray annotation for probes included sequential numbers added after bacteria genus or species in order to differentiate more than one probe with the same name. The fluorescence signal of each probe was read using a fluorescence array image reader (Sensovation AG, Radolfzell, Germany). Intensity was used as a parameter to determine semi quantitative relative fluorescence values for each probe, which were used to compare relative abundance of each microbiota taxa between feed groups according to the experimental design.

In addition, Pearson’s correlations were calculated between immune cells and intestinal microbial taxa that were significantly increased in each feed group. Positive and negative correlation values were reported between the microbial taxa mentioned in Table 2 and NK cell activation and subsets in the IELs and spleen for the respective intestinal segment, diet and age, and depicted in a heatmap. Correlation (r) values from 0 to 1 (positive) and 0 to −1 (negative) are shown, where 0–0.2 (0–−0.2) is interpreted as no/negligible correlation, 0.2–0.5 (−0.2–−0.5) as weak correlation, 0.5–0.8 (−0.5–−0.8) as moderate correlation, and 0.8–1 (−0.8–−1) as strong correlation. 

### 2.6. Statistical Analysis

First, normal distribution of the data was confirmed using the Shapiro–Wilk test. For the in vitro screening, differences in each of the assays between compounds and solvent controls were analyzed using one-way ANOVA. For the in ovo screening, differences in embryonic growth, hatchability, and white blood cell counts between feed groups were analyzed using a mixed-model ANOVA, where all variables were classed as fixed or random effects; diet was used as the fixed effect and block as the random effect. For the in vivo screening on performance traits, differences in performance traits between feed groups were analyzed using a mixed-model ANOVA, where diet was classed as the fixed effect and block as the random effect. For the in vivo experiment with glucose oligosaccharide and long-chain glucomannan, differences in numbers of IELs, splenic leukocytes, NK cell and T cell subsets, and percentages of CD107 expression in the IEL population and spleen between the feed groups were analyzed using one-way ANOVA. Regarding the analysis of microbiota composition, raw fluorescence intensity data for each probe on each microarray chip were compiled and submitted to data quality control. The selected data treatment to reduce chip-to-chip variation was to standardize it to the shifting point of three, according to the option available in JMP Genomics, as described previously for microarray analysis [59]. The standardized data were then analyzed using mixed-model ANOVA, where intestinal segment, diet, age, and their three-way interaction were classed as fixed effects and chip was classed as the random effect. Results were used to produce clustering plots utilizing hierarchical clustering, with distances between clusters defined by the Ward’s method [60]. Differences in standardized LS means were also used for principal component analysis and volcano plots for pair-wise comparisons. Correlations between immune cells and microbial taxa were analyzed using the Pearson product–moment correlation procedure. The statistical analyses for the in vitro screening and the immunological data of the in vivo experiment were performed using GraphPad Prism 9 software (GraphPad Software, San Diego, CA, USA). The statistical analyses for the in ovo and in vivo screening were performed using R software version 3.5.1 (The R Foundation for Statistical computing, Vienna, Austria) and for the microbiota data of the in vivo experiment using JMP Genomics 9 software (SAS Institute 2017, Cary, NC, USA). A *p* value of < 0.05 was considered statistically significant, a value of 0.05 < *p* < 0.1 was referred to as a trend. and in case the *p* value did not belong to one of these categories, it was referred to as a numerical difference. Microarray standardized LS means of fluorescence intensities were compared using false discovery rate (FDR) adjusted *p*-values set at < 0.05.

## 3. Results

### 3.1. Compounds Are Able to Induce Activation of NK Cells and Macrophages 

Sixty-nine compounds were screened in vitro for their capacity to induce enhanced CD107 expression on NK cells as well as to stimulate bead uptake and NO production by macrophage-like HD11 cells. Stimulation with 57 compounds resulted in enhanced surface expression of CD107 on NK cells compared to the solvent control (Appendix A). Viability of NK cells was not affected by stimulation with the compounds in the concentration range that was used (Figure 1a right panel). Stimulation with compounds P1 and P2 resulted in enhanced CD107 expression at all concentrations (Figure 1a, left panel), with the highest levels of CD107 expression at 10 ppm for both P1 (108.0 ± 7.1%) and P2 (103.3 ± 13.7%) compared to the solvent control (87.2 ± 0.8%). In contrast, stimulation with compound PE1 resulted in a lower CD107 expression at 50 and 100 ppm compared to the solvent control. Exposure of the macrophage cell line HD11 to 56 out of 69 compounds resulted in increased bead uptake compared to the solvent control (Appendix A). Stimulation with compound P1 resulted in increased bead uptake at 10 and 50 ppm, whereas increased bead uptake was observed upon stimulation with P2 at all concentrations compared to the control (Figure 1b, left panel). Phagocytoses were most pronounced with stimulation at 10 ppm for P1 (115.9 ± 4.3%) and P2 (116.0±2.2%) compared to the solvent control (86.3 ± 5.1%, Figure 1b, left panel). In contrast, stimulation with compound PE2 resulted in diminished bead uptake at all concentrations (Figure 1b, left panel). The viability of HD11 cells was not affected by P2, whereas exposure to P1 at 50 and 100 ppm and to PE2 (all concentrations) reduced viability of the cells (Figure 1b, right panel). Exposure to 22 out of 69 compounds increased NO production by macrophage-like HD11 cells compared to the solvent control (Appendix A). Stimulation with compound P1 increased NO production in all concentrations and highest at 50 ppm (96.8 ± 6.8 µM) compared to no NO production in the presence of the solvent control (Figure 1c). In contrast to P1, exposure to compound P2 did not result in NO production whereas stimulation with compound L1 led to low NO production at 10 ppm only (Figure 1c). Out of the compounds that showed positive effects on the activation of NK cells and macrophages, six compounds including P1, P2, P8, SS2, MS3, and L1 were selected and investigated further in the in ovo screening.

### 3.2. In Ovo Administration of the Selected Compounds Showed No Adverse Effects on Embryonic Development and Hatching 

Next, polysaccharides P1, P2, P8, simple sugar SS2, modified sugar MS3, and lipid L1 were injected into the amniotic fluid of ED18 chicken eggs and possible negative impact on embryonic growth, hatchability of eggs, and peripheral immune cell numbers in the chickens immediately post-hatch were determined. In ovo injection of compounds did not give rise to differences in relative yolk utilization and liver growth compared to the controls during embryonic development (Figure 2a). Injection of the compounds did not affect hatchability except for MS3 administered at a concentration of 2.0%, which decreased hatchability to 67.3 ± 13.4% compared to 100% of the controls (Figure 2b). No differences in numbers of lymphocytes, NK, B and T cells in the blood were observed in chicks from eggs that were injected with the compounds compared to chicks that hatched from eggs that received the solvent control (Figure 2c and Appendix A). Next, the effect of five out of these six compounds on performance traits was investigated. In this study, compound L1 was excluded since a practical route of administration to broiler chickens in vivo was not available.

### 3.3. No Negative Effects Were Observed on Performance Traits upon In Vivo Administration 

To determine possible negative effects on growth performance, diets supplemented with these five compounds were fed to broiler chickens from 0 days of age onwards. Two additional compounds, polysaccharide P5 and simple sugar SS2-S, were included in this study. These compounds are structurally related to P2 and SS2, respectively, and have been shown to enhance NK cell and macrophage activation in prior in vitro assays. No differences in body weight were observed among feed groups at 0, 7, 13, and 21 days of age (Appendix A). Furthermore, no differences in FCR as a result of the supplemented feeds compared to the standard diet (negative control) were observed (Appendix A). Comparisons of the feed additives with the positive control, which was feed supplemented with 0.0625% SCFP, showed that the FCR from 7 to 13 and 13 to 21 days of age of chickens in the P8 0.2% group was significantly higher compared to the positive control (1.34 ± 0.01 vs. 1.22 ± 0.01 and 1.28 ± 0.01 vs. 1.16 ± 0.01, respectively). A similar observation was made when comparing the P8 0.2% group with three other feed additives (SS2 0.2%; P1 0.02%; MS3 0.02%) during these age periods, while the FCR of chickens in the other groups were in between the FCR of P8 0.2% and the three other feed additives (Appendix A). None of the feed additives showed significant adverse effects on growth performance. P1 and P2 were the most favorable to select based on the numerical results of chicken body weight and FCR, and highest future potential feasibility. These two polysaccharides were studied in subsequent in vivo analyses for possible effects on immune cells and microbiota composition.

### 3.4. Glucose Oligosaccharide and Long-Chain Glucomannan Significantly Increase Activation of Intraepithelial NK Cells Seven Days after In Vivo Supplementation

Growth performance of broiler chickens was not affected by the administration of the supplemented feeds compared to the control diet (Appendix A). Furthermore, no significant differences were observed in numbers of IELs (Appendix A) and splenic leukocytes (Appendix A) of chickens in the supplemented feed groups compared to the standard diet group. Nevertheless, at 7 days of age, IELs were numerically higher in chickens that received feed containing long-chain glucomannan (F2, 8.3 × 10^4^ ± 2.6 × 10^4^) compared to the control group (4.4 × 10^4^ ± 6.0 × 10^3^, Appendix A). Next, NK cell subsets and NK cell activation were assessed in the IEL population and spleen (Figure 3a). At 7 days of age, intraepithelial IL-2Rα^+^ NK cells were numerically lower in chickens of the F1 group and 20E5^+^ NK cells were numerically higher in the F2 group compared to the control group, however, no significant differences were observed (Figure 3b,d). Significant enhanced CD107 expression on intraepithelial NK cells was observed at 7 days of age in both F1 and F2 groups, which was 21.3 ± 2.2% for F1 and 25.9 ± 1.7% for F2, compared to the control group (10.89 ± 1.1%, Figure 3f). In the spleen at day 14, IL-2Rα^+^ NK cells were numerically lower in the F1 group and 20E5^+^ NK cells were numerically higher in the F2 group compared to the control group, but no significant differences were observed (Figure 3c,e). Expression of CD107 on splenic NK cells was similar in chickens receiving one of the supplemented feeds compared to chickens that received a standard diet during aging (Figure 3g).

Furthermore, possible effects of the feed additives on the numbers of T cell subsets were analyzed (Figure 4a). Numbers of γδ T cells and CD8^+^ T cells were not significantly different between the F1 and F2 feed regimens compared to the standard diet in both the IEL population (Figure 4b,d) and spleen (Figure 4c,e). Nevertheless, numerical differences were observed at all ages in γδ T cells and CD8^+^ T cells in the F1 and F2 groups compared to the control group in IELs (Figure 4b,d) and spleen (Figure 4c,e). Likewise, no significant differences were observed in numbers of γδ T cells and CD8^+^ T cells expressing either CD8αα or CD8αβ (Appendix A). Expression of CD107 on intraepithelial CD8^+^ T cells (including both γδ and αβ T cells) was numerically increased in the F2 group at day 14 and 21 compared to the control group (Appendix A). In the spleen, CD107 expression on CD8^+^ T cells was numerically increased in the F2 group at 7 days of age compared to the control group (Appendix A). 

### 3.5. In Vivo Supplementation of Glucose Oligosaccharide and Long-Chain Glucomannan Led to Increased Relative Abundance of Lactobacillus Species in the Intestinal Microbiota 

Microbial analysis revealed that a total of 98 bacterial taxa identified by the probes were significantly different between the three feed groups among all tested factors (intestinal segment, age, diet, and their three-way interaction). By factorial analysis of the main effects, most of these significant differences were due to intestinal segment (78 bacterial taxa), followed by age (66 bacterial taxa), and diet (10 bacterial taxa). Principal component analysis (Figure 5) confirmed that microbiota profiles grouped first by ceca (left) and ileum (right) and subsequently by age (7, 14, and 21 days of age), however, grouping by age was affected by feed treatment (Control, F1, or F2). By factorial analysis of the three-way interaction; effects between feed groups at different ages in the ileum or ceca, F1 supplementation significantly increased the relative abundance of *Lactobacillus* species (*L. reuteri, L. crispatus*, and *L. gasseri*) in both ileum and ceca at 7 and 14 days of age, compared to the control group (2nd column Table 2, Figure 6 and Appendix A). The significantly increased intestinal bacteria of the F2 group compared to the control group were more variable in taxa in the intestinal segments and during aging, ranging from increased relative abundance of *L. crispatus* and *Bifidobacterium* in the ileum at 7 days of age and more proteolytic bacteria like *E. coli* and *Serratia marcescens* in the ceca at that same age, to increased relative abundance of *Lachnospiraceae* in the ceca at 14 days and *L. panis* in the ileum at 21 days of age (3rd column Table 2, Figure 6 and Appendix A). The significant differences in microbiota composition between F1 and F2 were more evident in chickens at 7 days of age, whereas microbiota profiles of chickens in both feed groups were more similar at days 14 and 21 (Table 2, 4th column, Figure 6 and Appendix A). At 7 days of age, F1 was associated with significantly higher relative abundance of *L. reuteri* in the ileum and *Ruminococcus* and *L. reuteri* in the ceca (4th column Table 2, Figure 6). In comparison, F2 was associated with significant higher relative abundance of *L. crispatus*, *L. jenseni*, and *Enterococcus* taxa in the ileum, while in the ceca to a variety of more strictly anaerobic bacteria such as *Faecalibacterium*, *Lachnospiraceae*, *D. formicigenerans*, *S. marcescens*, and *B. gallinarum* (4th column Table 2, Figure 6). In addition, the significantly increased relative abundance of *Lactobacillus* species in the ileum and ceca were higher in the F1 compared to F2 group at 7 days of age (Figure 6), 14, and 21 (Appendix A). Pathogenic bacteria were found only in the ceca at 14 days of age; significant higher relative abundance of *Salmonella* was observed in the intestinal microbiota of chickens in the control group and similar relative abundance of *C. jejuni* compared to the F2 group, although F2 was associated with higher relative abundance of *C. jejuni* compared to F1 (4th column Table 2, Appendix A).

### 3.6. Positive Correlations between NK Cell Activation and Lactic Acid Bacteria upon In Vivo Supplementation by Glucose Oligosaccharide and Long-Chain Glucomannan

As supplementation of the polysaccharides affected NK cells and microbiota composition, we analyzed whether these effects were related by performing a Pearson’s correlation analysis. At 7 days of age, strong positive correlations (0.8–1.0) were observed between CD107 expression on splenic NK cells and *Lactobacillus crispatus* 2 and 3 in the ileum microbiota of chickens in the control group (Figure 7a). In addition, strong negative correlations (−0.8–−1.0) were found between CD107 expression on splenic NK cells and *Lactobacillus* 3 and *Lactobacillus reuteri* 1, splenic IL-2Rα^+^ NK cell numbers, and *Clostridium bartletii* 1 and 2 as well as splenic 20E5^+^ NK cell numbers and *Lactobacillus jenseni* and *Lactobacillus crispatus* 3 in the ileum microbiota of chickens in the control group at day 7 (Figure 7a). In comparison, in the F1 group, strong positive correlations were observed between CD107 expression on intraepithelial NK cells and *Lactobacillus* 3 and *Clostridium bartletii* 2, intraepithelial IL-2Rα^+^ NK cell numbers, and *Clostridium bartletii* 1 as well as CD107 expression on splenic NK cells and *Bifidobacterium* 2 and *Lactobacillus crispatus* 3 (Figure 7a). In the F2 group, a strong negative correlation was found between intraepithelial IL-2Rα^+^ NK cell numbers and *Bifidobacterium* 2 and a strong positive correlation between intraepithelial 20E5^+^ NK cell numbers and *Clostridium bartletii* 2 in the ileum microbiota of chickens at 7 days of age (Figure 7a). In addition, strong positive correlations were observed between numbers of IL-2Rα^+^ NK cells in the spleen and *Enterococcus*, *Lactobacillus* 3, *Lactobacillus reuteri* 1, and *Lactobacillus crispatus* 3, whereas a strong negative correlation was seen with *Clostridium bartletii* 1 in the ileum microbiota of chickens in the F2 group at 7 days of age. Furthermore, strong positive correlations were observed between splenic 20E5^+^ NK cell numbers and *Enterococcus* and *Lactobacillus crispatus* 3 in the ileum microbiota of chickens in the F2 group at 7 days of age (Figure 7a). No, weak, or moderate correlations were observed between CD107 expression on NK cells and numbers of NK cell subsets in the IEL population and spleen and the relative abundance of microbial taxa due to the respective feed in the ceca at 7 days of age (Figure 7b) and in the ileum and ceca at 14 days of age (Appendix A) and 21 days of age (Appendix A). 

## 4. Discussion

Currently, feed additives supplemented to the diet to improve growth performance of broiler chickens with the additional benefit of modulation of immune responsiveness or intestinal microbiota composition are of high interest. 

The stimulatory effects of glucose oligosaccharide and long-chain glucomannan on the activation of NK cells and macrophages in vitro was in agreement with the reported immunomodulatory properties of plant-based polysaccharides observed in humans and chickens [32,45]. Furthermore, embryonic development, hatchability, and numbers of peripheral immune cells at day of hatch were not affected adversely by in ovo application of both polysaccharides, which indicates that they are safe for use as feed supplements. Moreover, this safety opens the option of in ovo application of both polysaccharides in addition to or instead of supplementation in the feed, to be investigated in future experiments. In the present study, in ovo application of these polysaccharides did neither affect the normal reduction of yolk sac reserves nor increase of liver weight [61], whereas a higher reduction would have suggested an improved metabolic rate of the developing embryo [60,62]. This may be due to the low digestibility of the polysaccharides in addition to the limited nutrient absorption capacity in chicken embryos due to lack of adequate nutrient transporters in the small intestine [63]. Both glucose oligosaccharide and long-chain glucomannan may be considered as prebiotics, which are undigestible materials consumed by gut microbiota [64], thereby modifying and selectively favoring beneficial microbes toward a healthier microbiota. 

Our finding that growth performance of broiler chickens was not affected by administration of both polysaccharides directly post-hatch was in agreement with other studies, although some prebiotics have shown beneficial effects on performance traits [65,66,67,68]. Supplementation with glucose oligosaccharide and long-chain glucomannan immediately post-hatch did increase intraepithelial NK cell activation early in life. This in vivo observation was in agreement with the enhanced NK cell activation observed in vitro and indicates direct immunomodulation by the polysaccharides as shown before with other polysaccharides [69]. In addition, the relation between the in vitro assays and in vivo situation suggests that these assays are useful tools to screen the immunomodulatory effects of feed constituents and may contribute to the reduction of animal experiments [70]. The enhanced activation of intraepithelial NK cells was observed after the first seven days of supplementation and subsequently decreased to levels similar to those observed in the control group. NK cells were shown to be involved in trained immunity in livestock [71] and humans [72]. Initial exposure to β-glucans and BCG vaccination induced responses of monocytes and NK cells, respectively, thereby priming these innate cells, and a subsequent exposure to bacterial components led to increased innate responses, conferring innate memory. In humans, trained NK cells have been shown upon a secondary stimulus to undergo a secondary expansion and have the capacity to more rapid degranulation and production of cytokines, resulting in a higher protective immunity status [72,73]. It could be hypothesized that the signs of early NK cell activation in our study are a consequence of training by the polysaccharides and that a secondary stimulus such as an infection could increase NK cell responses. Future studies should investigate whether these polysaccharides enhance NK cell (re-)activation in chickens in response to infections. In this study, no significant changes in numbers and activation of T cells in the IEL population and spleen were observed after supplementation, whereas other polysaccharides did increase percentages of intraepithelial T cells in chickens [74] and T cell activation in mammals [75,76]. This may be due to the different polysaccharides used in our study or that glucose oligosaccharide or long-chain glucomannan supplementation may have more pronounced effects on T cells during an infection, as was shown for another polysaccharide [77]. 

The microbiota compositions of the ileum and ceca were modulated by feed supplemented with either glucose oligosaccharide or long-chain glucomannan resulting in increased relative abundance of LAB such as *Lactobacillus* and *Bifidobacterium*, in agreement with other studies in broiler chickens [78,79]. This confirms that both the polymerized carbohydrate and the mannose polymer are fermented by LAB. The differences in relative abundance of intestinal microbial taxa were most evident at 7 days of age and subsequently decreased until 21 days of age, when, as shown previously [6], a mature stable microbiota composition has established in the intestine. Relative abundance of LAB was found to be higher in the case of glucose oligosaccharide supplementation compared to long-chain glucomannan supplementation, but the latter also stimulated relative abundance of more strictly anaerobic bacteria such as *Faecalibacterium* and *Lachnospiraceae*. In addition to being fermented by LAB, *Faecalibacterium*, and *Lachnospiraceae*, long-chain glucomannan may be fermented by a specific strain and, due to its microbial fermentation, result in metabolites to be utilized by other bacteria that increase in abundance [80,81]. Besides LAB, both *Faecalibacterium* [82] and *Lachnospiraceae* [83] have been shown to be beneficial for human intestinal health since intestinal and metabolic disorders were associated with depletion of these species. 

Then, we investigated whether the effects of supplemented feeds on NK cells and microbiota were related. Although similar analysis of correlation between immune cells and specific microbial taxa has been performed in mice [84], it has not been conducted in chickens. Both supplemented feeds enhanced CD107 expression on intraepithelial NK cells and increased relative abundance of commensal LAB at 7 days of age. Therefore, the positive correlations between NK cell activation and relative abundance of *L. reuteri* 1, *Lactobacillus* 3, *L. crispatus* 2, and *Bifidobacterium* 2 in the ileum at 7 days of age due to glucose oligosaccharide and long-chain glucomannan supplemented diets indicate involvement of LAB in the modulation of the function of NK cells. While glucose oligosaccharide showed positive correlations between relative abundance of LAB and NK cell subsets as well as activation of both intraepithelial and splenic NK cells, long-chain glucomannan mainly showed positive correlations with NK cells in the spleen. This suggests that glucose oligosaccharide affects NK cells both locally by interaction with LAB, and systemically by a yet hypothetical interaction with antigen presenting cells (APCs) that have interacted with intestinal LAB or translocation of microbial products into the circulation [85,86]. The effects of long-chain glucomannan are suggested to be more systemic rather than local according to the correlation analysis, where splenic NK cells are affected through the hypothetical mechanisms mentioned earlier. 

Enhancement of NK cell activation by microbiota has been shown previously in humans and mice [87], induced either directly or indirectly via APCs [88]. Intestinal APCs recognize microbial species through Toll-like receptors (TLRs), resulting in the production of cytokines such as interleukin-12 (IL-12), which induces NK as well as T cell responses such as IFNγ production [89,90]. It has been hypothesized that intraepithelial NK cells interact directly with microbiota by recognition of nonmethylated CpG motifs of bacterial DNA, mainly via TLR9, which enhances their cytotoxic activity [88,91]. TLR9 is absent in the chicken genome, however, TLR21 may act as a functional homologue to mammalian TLR9 in recognizing CPG motifs [92]. Furthermore, metabolites such as short-chain fatty acids are produced by microbiota and can be utilized as an energy substrate for microbial species as well as affect the function of intestinal epithelial cells, NK cells, APCs, and T cells [93,94]. These metabolites should be kept in mind when developing feed strategies to indirectly modulate the immune system via the microbiota [95]. Early feeding with glucose oligosaccharide and long-chain glucomannan may also improve health during adult life; either by preventing colonization of harmful bacteria or by the stimulation of the immune system and thereby increasing resistance against pathogens as previously shown for other dietary constituents in chickens [1,7] and humans [96]. 

In conclusion, this study showed that early feeding of glucose oligosaccharide and long-chain glucomannan stimulates intraepithelial NK cell activation as well as a relative abundance of commensal lactic acid bacteria in young broiler chickens. Although both feed additives had no effect on growth performance under non-challenging conditions, they may have an added value to performance by eliciting stronger immune responses during challenging conditions. Future studies should investigate the impact of feeding of these polysaccharides during experimental infection to validate their potency to improve resistance against bacteria and viruses in broiler chickens. 

## Figures and Tables

**Figure 1 vetsci-08-00110-f001:**
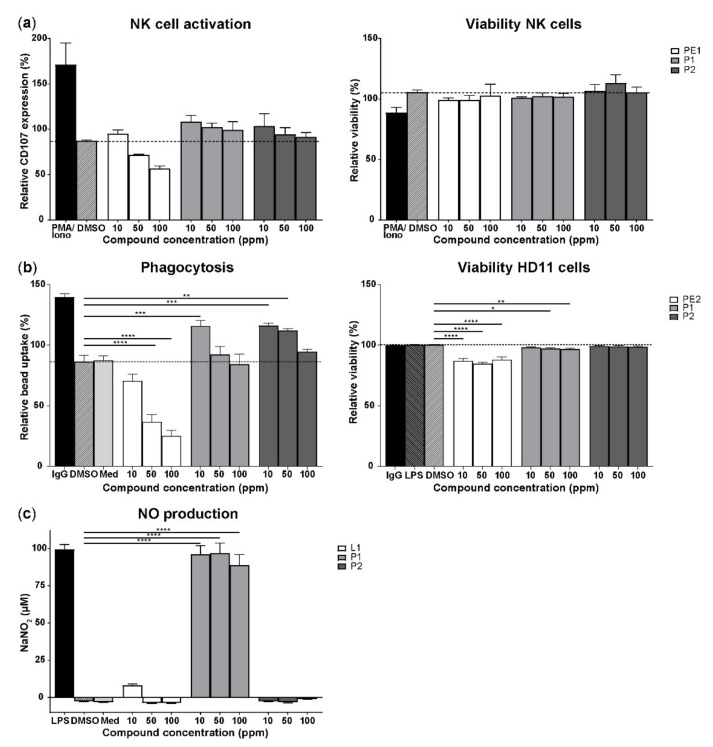
In vitro screening of feed compounds for their effect on the activation of NK cells and macrophages. (**a**) The effect of compounds on NK cell activation measured by the CD107 expression on NK cells (%). Expression of CD107 is expressed relative to the negative control, which was set at 100% (left panel). Viability of ED14 NK cells after exposure to compounds, which is expressed relative to the negative control set at 100% (right panel). PMA/Ionomycin was included as the positive control, 1:5 DMSO as the solvent control, and *n* = 3 per compound concentration. (**b**) The effect of compounds on the phagocytosis of macrophage-like HD11 cells (%). Phagocytosis is expressed relative to the reference control, which was set at 100% (left panel). Viability of HD11 cells after exposure to compounds, which is expressed relative to the negative control set at 100% (right panel). IgY-coated beads are included as positive control to determine the highest level of phagocytosis, LPS-coated beads in 1:5 DMSO as the solvent control, and uncoupled beads in complete RPMI medium as the negative control to determine baseline phagocytosis, *n* = 3 per compound concentration. (**c**) The effect of compounds on the NO production of macrophage-like HD11 cells by measuring nitrite concentration (µM). LPS stimulation was included as the positive control, 1:5 DMSO as the solvent control, complete RPMI medium as the negative control, and all samples were in triplicates, *n* = 9 per compound concentration. Mean + SEM is shown of the compounds of polysaccharides (P1, P2), plant extracts (PE1, PE2), and lipids (L1); the dotted line represents the level of the solvent control and statistical significance is indicated as * *p* < 0.05, ** *p* < 0.01, *** *p* < 0.001, and **** *p* < 0.0001.

**Figure 2 vetsci-08-00110-f002:**
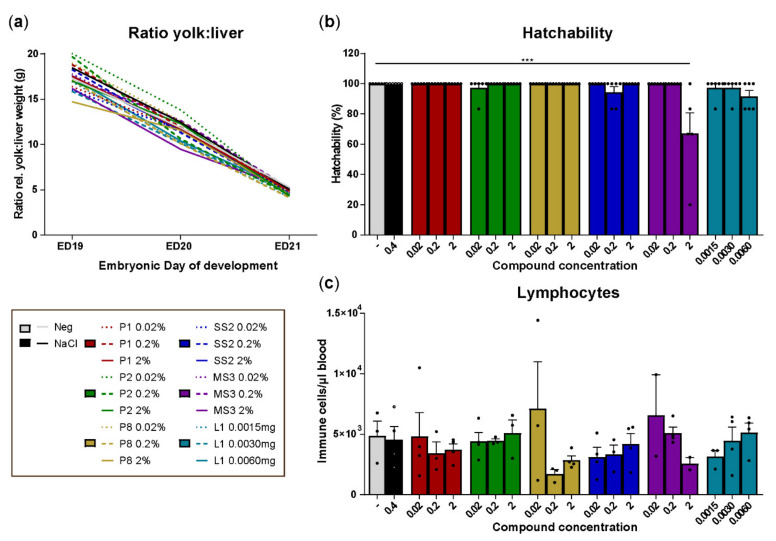
In ovo screening of feed compounds for their effect on embryonic growth, hatching, and peripheral lymphocyte numbers. (**a**) Effect of compound injections in embryos on the ratio of relative yolk and liver weight during late embryonic development, *n* = 6 per group per ED. (**b**) Effect of compound injections on hatchability (%) at ED21, *n* = 6 per group. (**c**) Effect of compound injections on the numbers of lymphocytes in whole blood in chicks immediately post-hatch. Non-injected eggs were included as the negative control and 0.4% saline solution as the solvent control, *n* = 4 per group. Mean + SEM is shown of compounds of polysaccharides (P1, P2, P8), simple sugars (SS2), modified sugars (MS3), and lipids (L1) in different concentrations (0.02, 0.2, 2%, or 0.0015, 0.0030, 0.0060 mg) and statistical significance is indicated as *** *p* < 0.001.

**Figure 3 vetsci-08-00110-f003:**
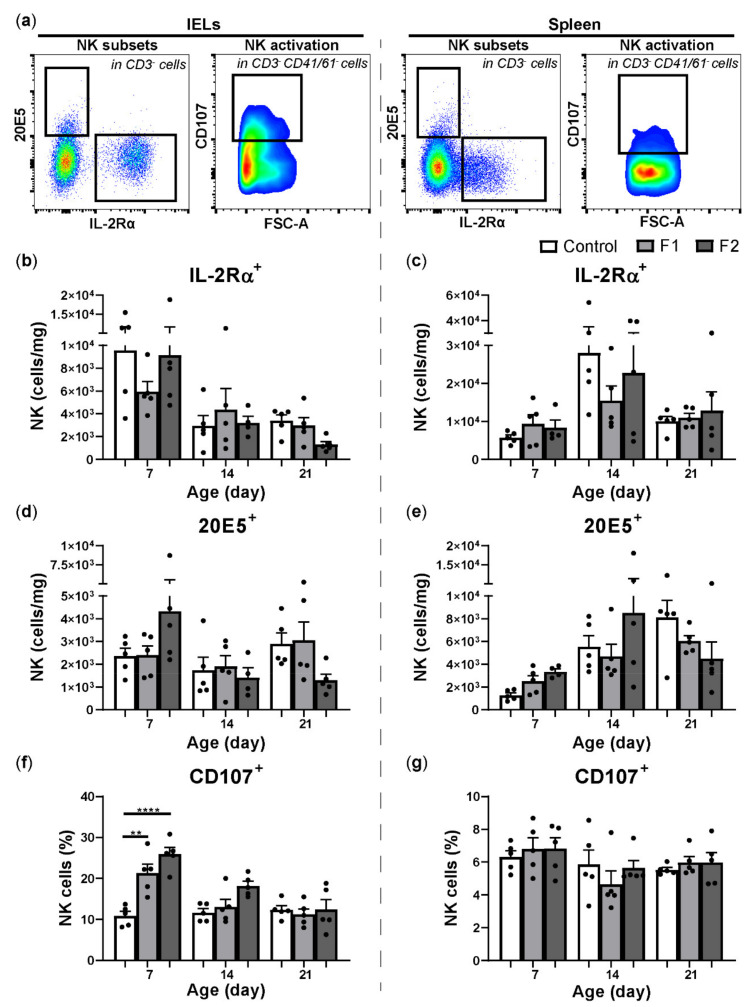
Effect of feeding glucose oligosaccharide and long-chain glucomannan to broiler chickens on NK cell subsets and NK cell activation. (**a**) CD3 negative cells expressing either IL-2Rα or 20E5, and CD3 and CD41/61 negative cells expressing CD107 in the IEL population (left panel) and spleen (right panel). (**b**) Numbers (cells/mg) of intraepithelial IL-2Rα^+^, (**d**) 20E5^+^ NK cells, and (**f**) percentages of NK cells expressing CD107 in broiler chickens provided different diets in the course of time. (**c**) Numbers (cells/mg) of splenic IL-2Rα^+^, (**e**) 20E5^+^ NK cells, and (**g**) percentages of NK cells expressing CD107 in broiler chickens provided different diets in the course of time. Mean + SEM is shown (*n* = 5) of chickens provided standard diet (control), feed supplemented with glucose oligosaccharide (F1) or long-chain glucomannan (F2) and statistical significance is indicated as ** *p* < 0.01 and **** *p* < 0.0001.

**Figure 4 vetsci-08-00110-f004:**
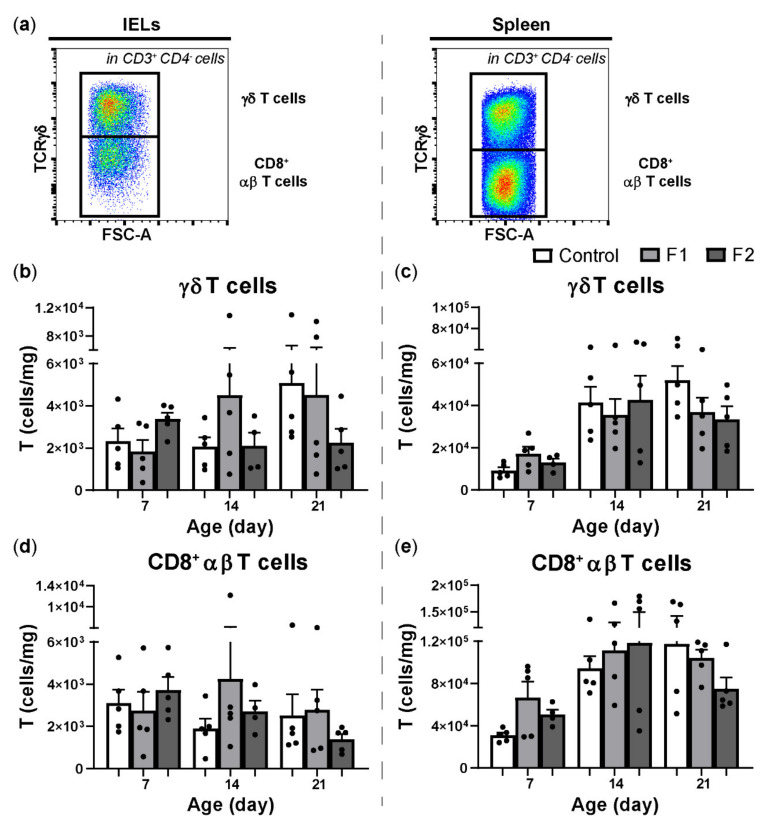
Effect of feeding glucose oligosaccharide and long-chain glucomannan to broiler chickens on T cell subsets. (**a**) CD3 positive and CD4 negative cells that are TCRγδ positive (γδ T cells) or negative (CD8^+^ αβ T cells) in the IEL population (left panel) and spleen (right panel). (**b**) Numbers (cells/mg) of intraepithelial γδ T cells and (**d**) CD8^+^ αβ T cells in broiler chickens fed with different diets in the course of time. (**c**) Numbers (cells/mg) of splenic γδ T cells and (**e**) CD8^+^ αβ T cells in broiler chickens fed with different diets in the course of time. Mean + SEM is shown (*n* = 5) of chickens provided standard diet (control), feed supplemented with glucose oligosaccharide (F1), or long-chain glucomannan (F2).

**Figure 5 vetsci-08-00110-f005:**
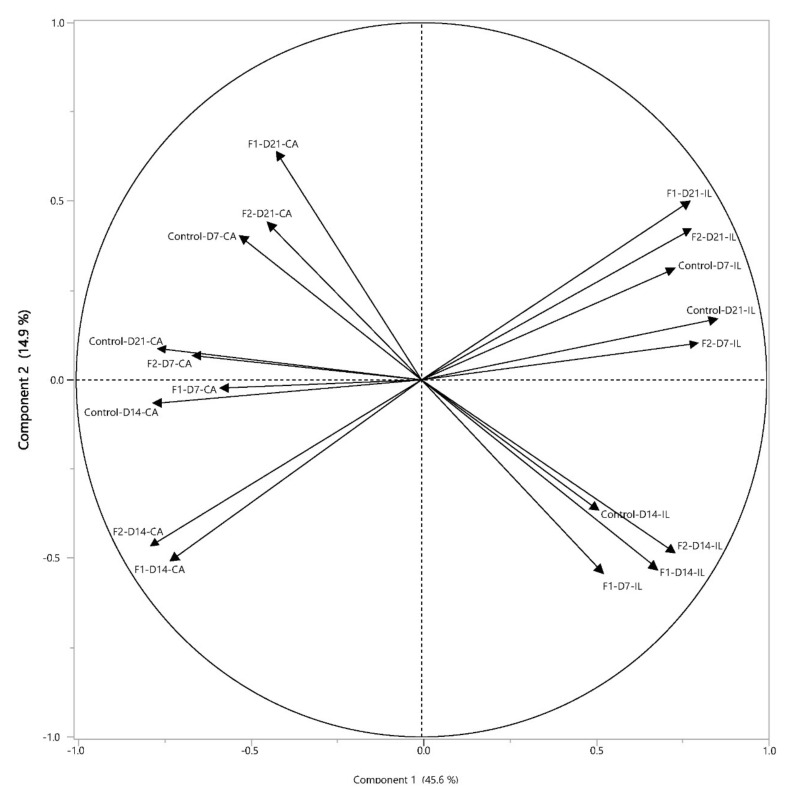
Effect of feeding glucose oligosaccharide and long-chain glucomannan to broiler chickens on microbiota composition in ileum and ceca. Principal component analysis of microbiota profiles from the ileum (IL) and ceca (CA) at day 7, 14, and 21 (D7, D14, and D21) of broiler chickens provided standard diet (control), feed supplemented with glucose oligosaccharide (F1), or long-chain glucomannan (F2). Per intestinal segment, age, and feed group *n* = 5.

**Figure 6 vetsci-08-00110-f006:**
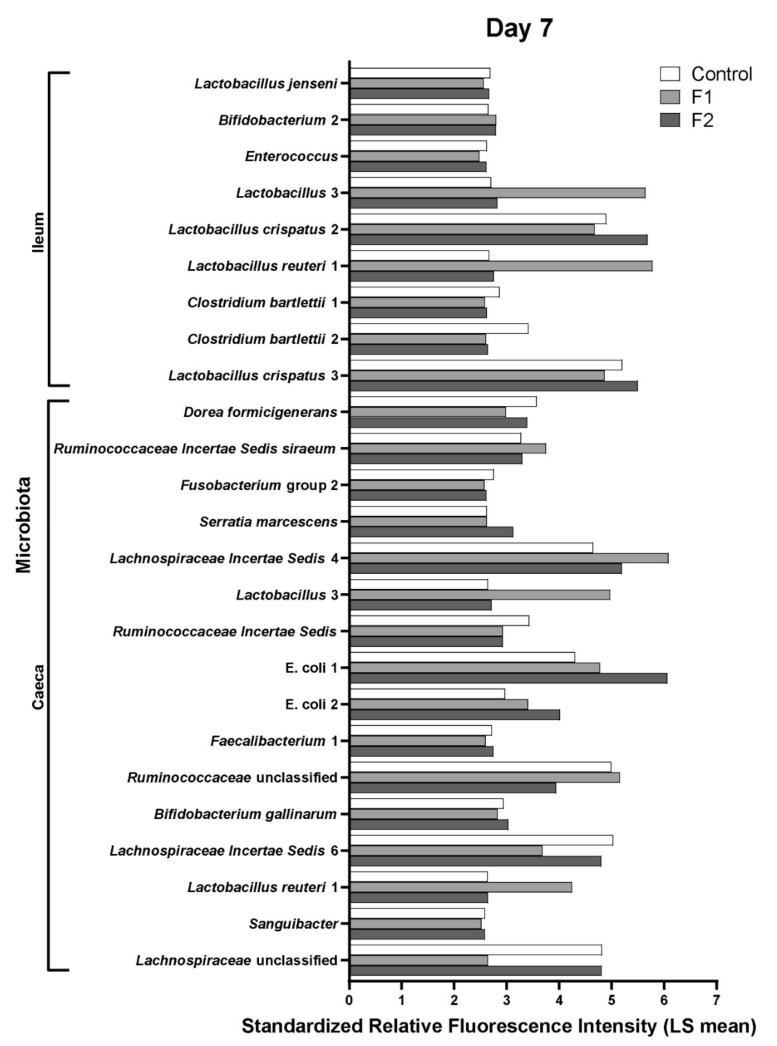
Relative abundance of intestinal microbial taxa significantly increased with the respective feed at day 7 in broiler chickens. Standardized relative fluorescence intensities (LS means) of the microbial taxa in the ileum and ceca as measured by the microarray (Table 2) that were significantly higher with standard diet (control), feed supplemented with glucose oligosaccharide (F1), or long-chain glucomannan (F2) at day 7 in broiler chickens. LS mean per microbial target and diet group are shown (*n* = 5) with statistical significance of FDR adjusted *p*-values set at < 0.05.

**Figure 7 vetsci-08-00110-f007:**
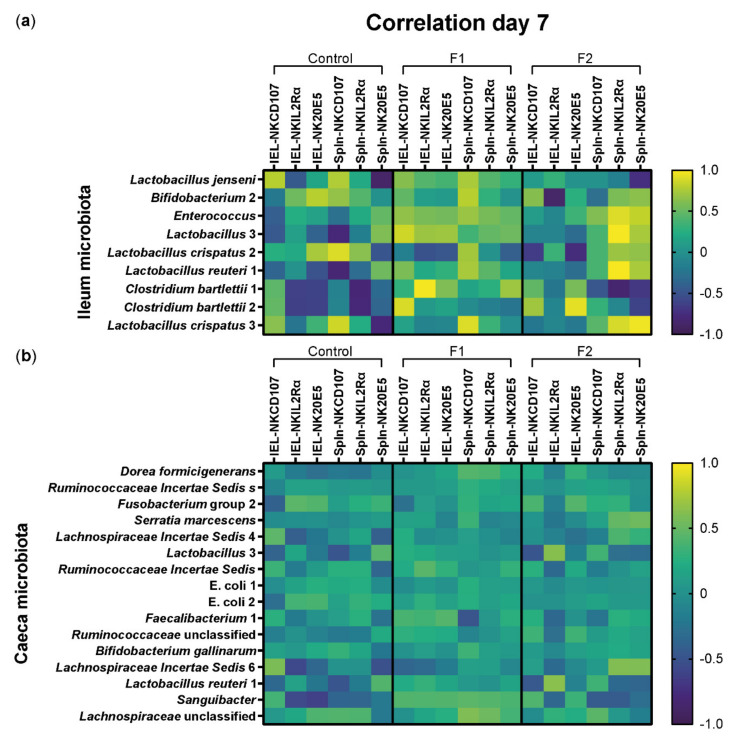
Correlations between intestinal microbial taxa and intraepithelial and splenic NK cells at day 7 in broiler chickens. (**a**) Correlation values between microbial taxa in the ileum or (**b**) ceca significantly increased with the respective feed and percentages of NK cell activation or numbers of NK cell subsets of the ileum (IELs) and spleen (Spln) per diet group (control, F1, F2) at day 7 in broiler chickens. Pearson’s correlation (r) values are depicted in a heatmap as positive (yellow) or negative (dark blue) correlations.

**Table 1 vetsci-08-00110-t001:** Flow cytometry staining reagents.

Cell Population	Primary Antibody (Mouse-Anti-Chicken)	Clone/Isotype	Secondary Antibody
Peripheral blood cells	CD45-PE ^1^	LT40/IgM	-
	Bu-1-FITC ^1^	AV20/IgG1	-
	CD3-PB ^1^	CT-3/IgG1	-
	CD4-APC ^1^	CT-4/IgG1	-
	CD8α-PE/Cy5 ^1^	CT-8/IgG1	-
NK cells	CD45-FITC ^1^	LT40/IgM	-
	CD3-APC ^1^	CT3/IgG1	-
	IL-2Rα-UNL ^2^	28–4/IgG3	Goat-anti-mouse-IgG3-PE ^1^
	20E5-BIOT ^2^	IgG1	Streptavidin (SA)-PercP ^5^
T cells	CD3-PE ^1^	CT3/IgG1	-
	CD4-APC ^1^	CT4/IgG1	-
	TCRγδ-FITC ^1^	TCR-1/IgG1	-
	CD8α-UNL ^1^	EP72/IgG2b	Goat-anti-mouse-IgG2b-APC/Cy7 ^1^
	CD8β-BIOT ^1^	EP42/IgG2a	SA-PercP^5^
Activation of NK and T cells			
in vitro CD107	CD107a-APC ^3^	LEP-100 I 5G10/IgG1	-
	CD41/61-FITC ^4^	11C3/IgG1	-
	CD3-PE ^1^	CT3/IgG1	-
in vivo CD107	CD107a-APC ^3^	LEP-100 I 5G10/IgG1	-
	CD41/61-FITC ^4^	11C3/IgG1	-
	CD3-PE ^1^	CT3/IgG1	-
	CD8α-UNL ^1^	EP72/IgG2b	Goat-anti-mouse-IgG2b-Alexa Fluor (AF) 790 ^6^

Manufacturer: ^1^ Southern Biotech, AL, USA, ^2^ Purified antibody from hybridoma supernatant kindly provided by Göbel, T.W., Ludwig Maximilian University, Germany, ^3^ Developmental Studies Hybridoma Bank (DSHB), University of Iowa, IA, USA, ^4^ Serotec, United Kingdom, ^5^ BD Biosciences, The Netherlands, ^6^ Biolegend, CA, USA.

**Table 2 vetsci-08-00110-t002:** Intestinal microbial taxa that are significantly increased with the respective feed at different ages in broiler chickens. The microbial taxa of which the standardized LS means were significantly increased due to the respective feed, as determined by factorial analysis of pairwise comparisons between feed groups at different ages in the ileum or the ceca. Feed groups included standard diet (control), and feed supplemented with glucose oligosaccharide (F1) or long-chain glucomannan (F2), with statistical significance of FDR adjusted *p*-values set at < 0.05.

Age/Intestinal Segment	Control vs. F1	Control vs. F2	F1 vs. F2
7 days/ileum	C: *Clostridium bartletii* 1 and 2, *Lactobacillus jenseni, Bifidobacterium* 2, *Enterococcus* sp.; F1: *Lactobacillus reuteri* 1, *Lactobacillus* 3;	C: *Clostridium bartletii* 1 and 2; F2: *Lactobacillus crispatus* 2, *Bifidobacterium* 2;	F1: *Lactobacillus* 3, *Lactobacillus reuteri* 1; F2: *Lactobacillus crispatus* 2, *Lactobacillus crispatus* 3, *Lactobacillus jenseni, Enterococcus* sp.;
7 days/ceca	C: *Lachnospiraceae* Incertae Sedis 6, *Dorea formicigenerans, Fusobacterium* group 2, *Sanguibacter*; F1: *Lactobacillus reuteri* 1, *Lactobacillus* 3, *Lachnospiraceae Incertae Sedis* 4, *Ruminococcus* sp.;	C: *Ruminococcus* sp.; F2: *E. coli* 1, *E. coli* 2, *Serratia marcescens*;	F1: *Ruminococcus* Incertae Sedis *siraeum, Ruminococcaceae* unclassified, *Lactobacillus* 3, *Lactobacillus reuteri* 1; F2: *Faecalibacterium* 1, *Lachnospiraceae* Incertae Sedis 6, *Lachnospiraceae* unclassified, *Dorea formicigenerans, Serratia marcescens, Bifidobacterium gallinarum*;
14 days/ileum	C: *Lachnospiraceae* Incertae Sedis 2, *Bacteroides* uncult, *Listeria*; F1: *Lactobacillus reuteri* 1, *Lactobacillus* 3;	C: *Lachnospiraceae* Incertae Sedis 2, *Bacteroides* uncult, *Clostridium bartletii* 2; F2: none	None
14 days/ceca	C: *Ruminococcus* unclassified, *Clostridiales* unclassified, Incertae Sedis Xlll unclassified; F1: *Lactobacillus* sp., *Lactobacillus gasseri* 2, *Lactobacillus crispatus* 3;	C: *Salmonella* 1, *Parabacteroides* 2, *Rikenellaceae Alistipes* 1, *Salinococcus*, *Clostridiales* unclassified, Incertae Sedis Xlll unclassified; F2: *Lachnospiraceae* Incertae Sedis 3;	F1: *Lachnospiraceae* Incertae Sedis 11; F2: *Campylobacter jejuni*;
21 days/ileum	None	C: *Lactobacillus crispatus* 2, *Bifidobacterium* 2, *Enterococcus hirae*; F2: *Lactobacillus panis*;	None
21 days/ceca	C: *Lachnospiraceae* unclassified, *Ruminococcus* Incertae Sedis; F1: none	C: *Lachnospiraceae* unclassified, *Ruminococcus* Incertae Sedis, *Citrobacter*; F2: Bacteria unclassified;	F1: *Agreia*; F2: none

## Data Availability

Data are contained within the article or Appendix A.

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
