# Peer review of "Glucose Oligosaccharide and Long-Chain Glucomannan Feed Additives Induce Enhanced Activation of Intraepithelial NK Cells and Relative Abundance of Commensal Lactic Acid Bacteria in Broiler Chickens"

_vetsci, 2021, doi:10.3390/vetsci8060110_

Round 1

Reviewer 1 Report

Excellent, well-written manuscript that provides a unique perspective on feed supplements effects on the innate immune response and microbiota.  For the most part the experimental design is excellent and  was conducted to answer the hypothesis presented.  I have a couple of questions/comments that I would like the authors to answer:

  1. Unless i missed it, how many times were the in vitro and in ovo screenings conducted before decisions were made as to the compounds to evaluate in vivo?  Further, how many times was the growth performance experiment conducted.  I realize that there were reps of the treatments within the experiment but was the actual experiment carried on at least once more wit a different hatch of birds?
  2. The phagocytosis screen using the HD11 cells provides interesting data on function of the macrophages, but phagocytosis without provide looking at killing seems to be an incomplete analysis of macrophage function.  The authors are aware that many bacterial pathogens are able to survive in macrophages regardless of active invasion or phagocytosis.  Simply evaluating phagocytosis of beads only tells half the story.  I would always prefer to see the whole story: do pathogens survive the phagocytosis or are they actually killed.  i realize that this may be somewhat of an overkill, but does provide a more sound basis for macrophage functional activity.
  3. Lastly, one of the issues in current poultry production is the development of a chronic, low grade intestinal inflammation induced by feed ingredients that stimulate the innate immune system in the gut.  As interesting as the compounds described here are, how do the authors perceive a prolonged feeding of such compounds and the potential for a chronic, low grade inflammation that keeps energy away from performance and towards maintenance of the energy-requiring innate immune response?

Reviewer 2 Report

See document attached

Reviewer 3 Report

The authors examined different compounds on the immune status and microbiota of broiler chicken. The manuscript is well-designed and well-written. 

The results of NK activation are somewhat exaggerated, as the differences were only noticed in Day-7. Please reduce the tone and correct the title accordingly.

The study also requires further in vivo and in vitro microbial challenge tests to examine the immune response to the pathogenic bacteria.

I have a minor comment:

L124: Please mention the names of the industrial collaborators in the text. I wonder if there is any conflict of interest?
